# Mechanistic Insights and Therapeutic Strategies in Osteoporosis: A Comprehensive Review

**DOI:** 10.3390/biomedicines12081635

**Published:** 2024-07-23

**Authors:** Nyruz Ramadan Elahmer, Sok Kuan Wong, Norazlina Mohamed, Ekram Alias, Kok-Yong Chin, Norliza Muhammad

**Affiliations:** 1Department of Pharmacology, Faculty of Medicine, Universiti Kebangsaan Malaysia, Kuala Lumpur 56000, Malaysia; nyruze@yahoo.com (N.R.E.); sokkuan@ukm.edu.my (S.K.W.); azlina@ppukm.ukm.edu.my (N.M.); chinkokyong@ppukm.ukm.edu.my (K.-Y.C.); 2Department of Pharmacology, Pharmacy Faculty, Elmergib University, Al Khums 40414, Libya; 3Department of Biochemistry, Faculty of Medicine, Universiti Kebangsaan Malaysia, Kuala Lumpur 56000, Malaysia; ekram.alias@ppukm.ukm.edu.my

**Keywords:** osteoporosis, bone remodeling, osteoblast, osteoclast, RANKL, Wnt, Notch receptors, cytokines

## Abstract

Osteoporosis, a metabolic bone disorder characterized by decreased bone mass per unit volume, poses a significant global health burden due to its association with heightened fracture risk and adverse impacts on patients’ quality of life. This review synthesizes the current understanding of the pathophysiological mechanisms underlying osteoporosis, with a focus on key regulatory pathways governing osteoblast and osteoclast activities. These pathways include RANK/RANKL/OPG, Wingless-int (Wnt)/β-catenin, and Jagged1/Notch1 signaling, alongside the involvement of parathyroid hormone (PTH) signaling, cytokine networks, and kynurenine in bone remodeling. Pharmacotherapeutic interventions targeting these pathways play a pivotal role in osteoporosis management. Anti-resorptive agents, such as bisphosphonates, estrogen replacement therapy/hormone replacement therapy (ERT/HRT), selective estrogen receptor modulators (SERMs), calcitonin, anti-RANKL antibodies, and cathepsin K inhibitors, aim to mitigate bone resorption. Conversely, anabolic agents, including PTH and anti-sclerostin drugs, stimulate bone formation. In addition to pharmacotherapy, nutritional supplementation with calcium, vitamin D, and vitamin K2 holds promise for osteoporosis prevention. However, despite the availability of therapeutic options, a substantial proportion of osteoporotic patients remain untreated, highlighting the need for improved clinical management strategies. This comprehensive review aims to provide clinicians and researchers with a mechanistic understanding of osteoporosis pathogenesis and the therapeutic mechanisms of existing medications. By elucidating these insights, this review seeks to inform evidence-based decision-making and optimize therapeutic outcomes for patients with osteoporosis.

## 1. Introduction

Osteoporosis stands as the most pervasive metabolic disorder, distinguished by diminished bone mass per unit volume, thereby elevating the risk of bone fractures and fragility. Its etiology involves diminished bone formation coupled with increased bone absorption, resulting in anomalous bone remodeling [1]. According to the World Health Organization (WHO), osteoporosis is diagnosed when the bone mineral density (BMD) is 2.5 standard deviations (SD) or more below the mean value for young, healthy women, denoted by a T-score of ≤2.5 SD [2]. Contributing factors to osteoporosis encompass aging, familial predisposition, ethnic background, deficient dietary intake of vitamin D and calcium, immobility, menopause, excessive alcohol consumption, certain medications (e.g., anticonvulsants and glucocorticoids), hyperthyroidism, anorexia nervosa, Cushing’s disease, and inflammation [3]. Both genders are susceptible to osteoporosis across all age groups, with postmenopausal women exhibiting a heightened vulnerability [4]. The incidence of osteoporosis is escalating precipitously, correlating with the aging demographic [5]. Annually, osteoporosis impacts an estimated 75 million individuals globally [6]. Forecasts indicate that by 2050, Asia is anticipated to exhibit the highest prevalence of osteoporosis, accounting for 50% of all osteoporotic fractures worldwide [7]. Remarkably, the economic burden associated with osteoporosis management is substantial, with projected costs of USD 13.7–20.3 billion in the United States [8], EUR 31 billion in the European Union [9], and USD 13 billion in the Asia Pacific region [10]. These costs encompass various aspects of osteoporosis care, including diagnostics, pharmacotherapy, hospitalization, and rehabilitation. Medications such as denosumab, an anti-resorptive agent, contribute to these economic implications, as they are associated with significant costs for both patients and healthcare systems [11]. 

Osteoporosis is classified into primary and secondary forms. Primary osteoporosis further delineates into type I osteoporosis (postmenopausal osteoporosis), characterized by a high turnover rate, and type II (senile osteoporosis), which manifests with a low turnover rate. The elevated risk of primary type I osteoporosis in women is attributed to diminished estrogen levels and accelerated bone resorption following menopause [12,13]. Conversely, primary type II osteoporosis, affecting elderly individuals of both genders, is primarily a consequence of the aging process. Secondary osteoporosis is often associated with underlying diseases and/or medication usage [14]. Contributing factors to secondary osteoporosis encompass the intake of certain drugs such as anticonvulsants, heparin, thiazolidinediones, glucocorticoids, and immunosuppressants, as well as oncology therapies. Chronic conditions, including liver cirrhosis, renal impairment, malabsorption, and endocrine disorders such as Cushing’s syndrome, thyrotoxicosis, hypogonadism, and hyperparathyroidism are also implicated [15].

While numerous clinical trials substantiate the efficacy of these treatments, their extended usage is often associated with side effects [16]. This review aims to consolidate mechanistic insights into osteoporosis and the current pharmacological landscape, providing clinicians with valuable information for optimal therapeutic decision-making. In conclusion, this comprehensive review aspires to illuminate the intricate mechanistic underpinnings of osteoporosis, delving into the molecular and cellular processes that contribute to its pathogenesis. Additionally, a thorough exploration of the current armamentarium of treatments for osteoporosis is presented, elucidating the mechanisms of action employed by these therapeutic modalities. Furthermore, an appraisal of the documented side effects associated with prolonged treatment regimens is provided, affording clinicians and researchers a holistic understanding of the nuanced interplay between osteoporotic pathophysiology, therapeutic interventions, and their potential drawbacks. By amalgamating these critical facets, this review aims to serve as a valuable resource, fostering an enhanced comprehension of osteoporosis for the scientific community and, importantly, aiding clinicians in formulating informed decisions on the management of this prevalent and clinically significant disorder.

## 2. Bone Remodeling

Bone remodeling is an ongoing cycle of formation and degradation that provides bones with their shape and preserves calcium levels in the bloodstream at a healthy level. The degradation of bone is performed by giant cells known as osteoclasts to release calcium to support the body’s metabolic requirements. In addition, osteoclasts enable the bone to change in size and shape as it develops to adult dimensions. Osteoclasts degrade bone at different locations, whereas osteoblasts create new bone to preserve the skeletal structure. During childhood, bone production outperforms bone destruction as development continues. After reaching physiological maturity, the two processes remain at an equal balance [17].

Activation, resorption, reversal, formation, mineralization, and termination are the six successive steps of the remodeling cycle. The initial stage of bone remodeling requires the identification of an initiating remodeling indication. Then, during the resorption stage, osteoblasts react to osteocyte-generated signaling or direct endocrine stimulation signaling, attracting osteoclast precursors to the remodeling region. Interestingly, the resorption phase lasts only a limited time, depending on the intensity of the triggers that cause the differentiation and the activation of osteoclasts. The resorption stage is followed by the reversal stage, which is distinguished by the elimination of nearly all osteoclasts. Then, the total replacing of osteoclasts with osteoblasts distinguishes the formation stage. Finally, the osteoblasts’ final differentiation is one of the signs that bone remodeling has terminated. Up until the start of the subsequent wave of remodeling, the surface micro-environment of the resting bone is preserved [18].

### 2.1. Cells Involved in Bone Remodeling

The bone remodeling process is controlled by several cell types: osteoblasts, which facilitate bone formation; osteocytes, which maintain bone homeostasis; osteoclasts, responsible for bone resorption; and immune cells, both innate and adaptive, that regulate both bone formation and resorption [19].

At the forefront of bone remodeling are osteoblasts, the cells responsible for synthesizing and depositing new bone tissue. Derived from mesenchymal stem cells, osteoblasts possess the remarkable ability to produce and organize the extracellular matrix (ECM) that forms bone. They secrete collagen, the primary structural protein in bone, and promote the mineralization of the ECM with calcium and phosphate, resulting in the hard, calcified bone matrix. Osteoblasts also produce various growth factors, including transforming growth factor-beta (TGF-β) and insulin-like growth factors (IGFs), which stimulate bone formation [20]. Osteoblast differentiation is regulated by Runx2 (runt-related transcription factor 2) and other transcription factors [19]. A subset of osteoblasts that undergo final differentiation and are absorbed by osteoid during bone formation is known as osteocytes [17].

Nestled within the bone matrix, osteocytes are considered the master regulators of bone remodeling. Once osteoblasts become surrounded by mineralized ECM, they differentiate into osteocytes. Osteocytes form an intricate network of communication channels known as canaliculi, enabling them to sense mechanical strain and chemical signals. Through this network, osteocytes orchestrate the activities of osteoblasts and osteoclasts, responding to both systemic and local factors. When under mechanical stress, they produce molecules such as sclerostin, which inhibits bone formation by blocking Wingless-int (Wnt) signaling. Parathyroid hormone signaling inhibits sclerostin expression to unblock Wnt signaling and enable Wnt-directed bone formation. Both osteocytes and osteoblasts release osteoclastogenic factors such as receptor activators of the NF-κB ligand (RANKL) and macrophage colony-stimulating factor-1 (CSF-1) in response to stimulation to cause bone remodeling [21]. Osteocytes also participate in calcium homeostasis, ensuring optimal mineralization within the bone tissue [22].

While osteoblasts build bone, osteoclasts perform the essential task of bone resorption. These multinucleated cells, derived from the monocyte/macrophage lineage, possess a unique ability to degrade bone tissue. They are located in resorption bays, often referred to as Howship’s Lacunae. Osteoclasts secrete enzymes, such as tartrate-resistant acid phosphatase (TRAP) and cathepsin K, which break down the mineralized matrix, liberating calcium and phosphate into the bloodstream. This process is vital for calcium homeostasis, as well as the removal of damaged or aged bone. Osteoclasts are regulated by various factors, including RANKL and macrophage colony-stimulating factor (M-CSF), which initiates the formation and activation of osteoclasts [23]. 

The fourth type of cells is the innate immune cells, primarily polymorphonuclear neutrophils (PMNS) and monocytes/macrophages. Along with adaptive immune cells, they influence bone resorption during inflammatory responses. PMNS produce membrane-bound RANKL and inflammatory cytokines such as TNF-α, IL-1β, and IL-6. M1 and M2 macrophages are inflammatory phenotypes [24]. M1 releases cytokines, including IL-1β, TNF-α, and RANKL, whereas M2 releases anti-inflammatory mediators such as IL-4 and IL-10. Activated B and T cells produce cytokines and RANKL, which generally stimulate osteoclastogenesis. Interestingly, T cells can differentiate into Th1, Th2, and Th7. While Th1 expresses TNF-α and IL-1, which stimulate bone resorption, Th17 expresses pro-inflammatory cytokines IL-17 and IL-1, which promote osteoclastogenesis via RANKL induction. However, Th2 releases anti-inflammatory cytokines (IL-10 and IL-4) that inhibit osteoclastogenesis. Significantly, B cells produce IL-6, TNF-α, and RANKL, which induce osteoclastogenesis [25], (Figure 1).

### 2.2. Regulation of Bone Remodeling

There are crucial pathways that regulate osteoblasts and osteoclasts activities to influence bone mass density, including RANK/RANKL/OPG, Wnt/β-catenin, and Jagged1/Notch1 signaling [26]. Parathyroid hormone PTH signaling, cytokines complex network, and kynurenine are also involved in bone remodeling [27,28,29]. 

#### 2.2.1. RANK/RANKL/OPG Signaling Pathway

NFAT has an important stimulatory function in osteoblast proliferation and differentiation, which results in the production of non-collagenous and collagen proteins. Runx2, a master transcription factor for osteoblast development, is initially expressed in progenitor cells, resulting in the production of preosteoblasts [30]. Runx2 activates Sp7 (Osterix) in preosteoblasts, resulting in the start of mineralization and the creation of an extracellular matrix [31]. Consequently, mature osteoblasts often express a high level of bone gamma carboxyglutamate protein, including Bglap and osteocalcin [31]. Dentin matrix protein 1 (Dmp1), fibroblast growth factor 23 (Fgf23), and sclerostin (Sost) expression are markers of the osteocyte phenotype, which is reached by the osteoblasts when they are surrounded by mineralized bone [31]. Osteocytes, which make up 90% of all bone cells, play a role in maintaining bone homeostasis since they are the main source of the cytokine receptor activator of RANKL and Tnfsf11, which triggers osteoclastogenesis on osteoclast progenitors [32]. RANKL is also expressed by osteoblast lineage cells [32]. RANKL binds to the osteoclast surface via the RANK receptor to produce its osteoresorptive actions. Notably, osteoprotegerin is expressed by osteoblastic stromal cells that prevent RANKL from attaching to RANK, thus suppressing osteoclast development and maturation. As a result, activated osteoclasts adhere to the surface of the bone, release proteinase and protons, dissolve the minerals in the bone, and break down the matrix [33]. The two primary proteinases thought to be involved in the solubilization of collagenous matrix are cysteine proteinase and matrix metalloproteinase (MMP) families. Even though MMP-9, which is not rate-restricted, is the most common gelatinolytic MMP in osteoclasts, the key MMPs responsible for the destruction of bone collagen remain. On the one hand, TIMP-1 and TIMP-2 reduce the effects of MMP-9 and -2 via the zinc-dependent endopeptidase actions, respectively [34]. On the other hand, the action of cysteine proteinase on matrix solubilization is increased by cathepsin K, which is rate-restricted [35]. Interestingly, cathepsin K can break the collagen triple helix at many places, enhancing the vulnerability of the triple helix to any proteinase [35]. Cathepsin K has the highest gelatinolytic activity of all the cysteine proteinases [35]. MMPs and cathepsin K are the important proteinases in the procedure as a result [34,35], (Figure 2).

Various medical agents specifically target the RANK/RANKL/OPG signaling pathway to treat osteoporosis. Anti-resorptive drugs, such as bisphosphonates, reduce bone resorption by impairing osteoclasts’ ability to form ruffled borders and release protons necessary for bone resorption [36]. Similarly, estrogen replacement therapy (ERT)/hormone replacement therapy (HRT) decreases RANKL levels and promotes the synthesis of OPG [37]. Selective estrogen receptor modulators (SERMs) work through a mechanism similar to that of ERT [38]. Additionally, anti-RANKL antibodies act as RANKL antagonists, and cathepsin K inhibitors also play a role in this therapeutic category. Moreover, anabolic drugs like strontium ranelate, which was discontinued for commercial reasons [39], and prophylactic agents such as vitamin K2 are also utilized [40].

#### 2.2.2. Wnt/β-Catenin Signaling Pathway or Canonical Pathway

Several distinct cellular differentiations are significantly regulated by the Wnt signaling pathway [41]. The Wnt pathway has a significant impact on all stages of skeletogenesis, including patterning of embryonic skeletal, development of fetal skeletal, and remodeling of adult bone [41]. Wnt/β-catenin signal transduction has downstream factors, including Frizzled-2, Runx2, Axin 2, and β-catenin in osteoblasts, where they particularly induce osteoblast differentiation and mineralization [42]. Various human Wnt proteins have been identified. When Wnt glycoproteins bind to the frizzled receptor (Fz) and a co-receptor called low-density lipoprotein receptor-related protein LRP 5 or 6, they initiate Wnt/β-catenin signaling [43]. Followed by Axin2 stimulation by glycogen synthase kinase 3β (GSK3β), this process prevents β-catenin phosphorylation [28]. Consequently, β-catenin is prevented from being degraded, which causes it to accumulate, be transported into the nucleus, and regulate several target genes expression via interacting with different transcription factors such as TBX5, TCF/LEF, and HIF-1α23 [42]. β-catenin suppresses osteoclastogenesis by boosting OPG synthesis in osteoblasts and controlling the RANK/RANKL/OPG signaling. As a result, stimulating the canonical Wnt signaling improves bone production while decreasing bone resorption. Therefore, Wnt/β-catenin signals induce the proliferation and differentiation of osteoblasts in a β-catenin-dependent or independent way [42]. However, the amount of β-catenin is kept low in the absence of a Wnt signal via the breakdown of cytoplasmic β-catenin [44]. By phosphorylating certain amino acid residues in β-catenin, a multiprotein complex made up of the scaffolding proteins axin, GSK3β and adenomatosis polyposis coli (APC) mediates β-catenin degradation [44]. Various factors such as Wnt inhibitory factor (WIF) and secreted frizzled-related proteins (SFRPs) interact with Wnt proteins, preventing activation of the frizzled receptor (Fz) and a co-receptor LRP 5 or 6 [45]. Moreover, endogenous factors such as proteins of the DKK1 family and sclerostin competitively bind with LRP5/6, leading to Wnt signaling inhibition and bone formation suppression [45], (Figure 3).

Significantly, the Wnt/β-catenin signaling pathway is considered a therapeutic target to treat osteoporosis. The medical treatments include ERT/HRT, which acts on the Wnt/β-catenin signaling pathway and stimulates progenitor cell proliferation and differentiation. Also, calcitonin, which stimulates Wnt10b in osteoclasts, has a medical effect on the Wnt/β-catenin signaling pathway. Anabolic drugs such as anti-sclerostin antibodies are considered medical agents, acting on the Wnt/β-catenin signaling pathway.

#### 2.2.3. Jagged1/Notch1 Signaling

In osteoblasts, the Jagged1/Notch1 pathway adversely mediates osteoclastogenesis indirectly by changing the RANKL/OPG expression ratio in stromal cells and directly by reducing the number of osteoclast progenitors [26]. The extracellular delta, serrate, lag-2 consensus sequence (DSL) domain of the 180 kDa type I transmembrane glycoprotein known as Jagged1 is required for the interaction of Jagged1 with Notch receptors [46]. Jagged1/Notch1 signaling regulates cell proliferation and differentiation and defines cell fate, while direct cell–cell contact is believed to be essential to activate Notch signaling [46], (Figure 4).

However, currently, there are no available treatments that target Jagged1/Notch1 signaling pathway. Consequently, further studies are required to investigate medical agents that might have a role in Jagged1/Notch1 signaling pathway in treating osteoporosis.

#### 2.2.4. Parathyroid Hormone PTH Signaling

Endogenous PTH secretion typically increases over the lifespan. Normally, PTH enhances calcium absorption from the gut and reabsorption in the renal tubules in response to decreased blood calcium levels. Additionally, it stimulates calcium release from bone through bone resorption [47]. This signaling is initiated when PTH binds to the osteoblastic PTH receptor (PTH1R), leading to increased RANKL expression and decreased OPG expression. As a result, osteoclastogenesis and bone resorption are accelerated [47] (Figure 5). However, PTH also exhibits an anabolic effect on bone remodeling by stimulating osteoblast proliferation and differentiation. Notably, PTH reduces osteoblast apoptosis and promotes the secretion of Wnt from osteocytes [48,49], (Figure 5). Conversely, chronic elevation of PTH in old age typically results in bone resorption. Therefore, intermittent dosing of PTH generally has an overall anabolic effect on bone, making it a viable therapeutic option [28]. As a therapeutic target for osteoporosis, PTH is considered an anabolic drug. 

#### 2.2.5. Pro-Inflammatory Cytokines Complex Network

Bone remodeling is mediated via the cytokines complex network. Osteoclastogenic cytokines, including IL-31, IL-6, IL-17, TNF-α, and IFN-γ, suppress osteoblast proliferation and differentiation. Meanwhile, other cytokines, including IL-10, IL-12, IL-4, and IL-33, suppress osteoclastogenesis and stimulate the function of osteogenetics [50]. Th1 and Th17 cell cytokines are osteoclastogenic, while Th2 cell cytokines such as IL-33, IL-10, and IL-4 are anti-osteoclastic [51].

Recent research has shown that postmenopausal women who have estrogen deficiency with osteoporosis have low blood levels of IL-33 [52]. IL-33 has anti-osteoclastic properties, leading to the potential use of recombinant IL-33 in the treatment of osteoporosis [51]. IL-4 also decreases osteoclast growth [51] and RANKL expression in the bone marrow and RAW264.7 cells, leading to decreased bone resorption [53]. Similarly, IL-10 inhibits osteoclast formation, suggesting IL-10’s role in bone mass loss [54]. On the contrary, Th17 differentiation, which secretes IL-17, is enhanced with estrogen deficiency, and IL-17 stimulates osteoclastogenesis [55]. Consequently, the deletion of IL-17RA, the main IL-17 receptor, inhibited bone loss in the ovariectomized rat [56]. Similarly, IL-6 is enhanced with estrogen deficiency, and it increases osteoclastogenesis [57]. Meanwhile, TNF-α stimulates osteoclastogenesis directly by activating mature osteoclasts and indirectly via increases in RANKL secretion from stromal cells and osteoblasts in case of estrogen deficiency [58]. IFN-γ has a paradoxical effect on osteoclast formation. IFN-γ decreases osteoclastogenesis directly in vitro [59] and increases osteoclastogenesis indirectly in vivo [51]. Significantly, cytokines inhibitors can be used to treat osteoporosis, such as TNF-α inhibitors, IL-6 inhibitors, and IL-17 inhibitors.

#### 2.2.6. Kynurenine (KYN) Pathway

The KYN pathway is a novel treatment target for osteoporosis [29]. It stimulates different associated genes to break down tryptophan [60]. The KYN pathway metabolites accumulate with age and affect the differentiation of osteoprogenitor cells to osteoblast cells, leading to an increase in bone resorption markers such as TRAP-5b and RANKL but not OPG and BSALP and bone fragility in older adults [61]. On the other hand, picolinic acid, which is a by-product of the KYN pathway, stimulates bone formation in mice and has a potent osteogenic impact on MSCs [60], (Figure 6). Therefore, understanding the precise mechanism by which tryptophan metabolites affect bone cells may help in the creation of potent drugs that have both an anti-resorptive and anabolic impact on the bone.

## 3. Current Pharmacological Treatment

Several drugs have shown their benefits in enhancing BMD and minimizing the risk of bone fractures by increasing bone formation or slowing bone resorption. Osteoporosis treatments include anti-resorption medications, anabolic medications, and prophylactic therapies based on necessary nutrients such as vitamin D, calcium, and vitamin K2 [16].

### 3.1. Anti-Resorptive Agents

Anti-resorptive agents, which decrease bone resorption, include bisphosphonates, ERT/HRT therapy and SERMs, calcitonin, anti-RANKL antibody, and cathepsin K inhibitors [16].

#### 3.1.1. Bisphosphonates

Bisphosphonates, the anti-resorptive drugs, such as ibandronate, alendronate, zoledronic acid (ZA), and risedronate, are classified as the first-line therapeutic interventions for osteoporosis [62]. Bisphosphonates are pyrophosphate analogs with a P−C−P bond that offers the binding ability to hydroxyapatite on bone surfaces [62]. While osteoclasts resorb bone matrix, bisphosphonates decrease osteoclasts’ ability to create ruffled borders and release the protons required for bone resorption. Bisphosphonates also induce osteoclast apoptosis, leading to a decrease in bone loss and an increase in BMD [36]. The chemical structure of bisphosphonates significantly influences their anti-resorptive activity. Each molecule consists of two pyrophosphate groups, with variations in the R1 group that can be chlorine (Cl), hydrogen (H), or hydroxyl (OH). The hydroxyl group on the central carbon of bisphosphonates serves as an effective anchor to the bone matrix, enhancing the molecule’s selectivity for bone [63]. Additionally, the R2 group attached to the central carbon is crucial for anti-resorptive efficacy. Notably, nitrogen-containing R2 groups can increase the anti-resorptive activity of bisphosphonates by a factor ranging from 10 to 10,000 [64], (Figure 7).

Bisphosphonates are categorized into two types: the non-nitrogen-containing group and the nitrogen-containing group. Non-nitrogen-containing group, including clodronate, etidronate, and tiludronate (Figure 8), inhibits mitochondrial ATP translocases in osteoclasts and causes osteoclasts apoptosis [65]. On the other hand, the nitrogen-containing group, including alendronate, risedronate, zoledronate, and ibandronate (Figure 8), affects farnesyl diphosphate (FPP) synthase in osteoclasts, obstructs protein prenylation that is required for osteoclasts survival and function and decreases bone resorption [64] (Figure 9). Interestingly, the nitrogen-containing group is more effective than the non-nitrogen-containing group [64]. As there are various types of nitrogen-containing bisphosphonates, their abilities to bind bone surfaces and therapeutical potency are different. The ranking of bisphosphonates’ binding ability is as follows: zoledronate > alendronate > ibandronate > risedronate [66]. Bisphosphonates that have a higher binding ability move slowly throughout the bone and have limited network access to osteocytes [66]. On the other hand, bisphosphonates that have a lower binding ability spread widely through the bone and possess a short residence time after therapy is discontinued, compared with higher binding ability agents [66]. Recently, the FDA-approved bisphosphonates for osteoporosis, including zoledronate, alendronate, ibandronate, and risedronate, are all nitrogen-containing agents [67].

Alendronate is among the most widely used bisphosphonates, and the FDA has certified it for postmenopausal osteoporosis prevention and treatment. Indeed, alendronate is certified for boosting bone mass in osteoporotic men. Significantly, it improves BMD and reduces the risk of fractures of both the hip and the spine [68]. In addition, ibandronate is approved to treat postmenopausal osteoporosis. It decreases vertebral fractures, but it does not have any effect on nonvertebral fractures [68]. Risedronate is FDA-approved as a prevention and treatment of osteoporosis for postmenopausal women. It decreases vertebral and nonvertebral fractures [68]. Zoledronate is also FDA-approved as a prevention and treatment of osteoporosis for postmenopausal women. Zoledronate is different from the other nitrogen-containing agents as it has two atoms of nitrogen in a ring of heterocyclic imidazole [68]. It decreases 70% of vertebral fractures, 41% of hip fractures, and 25% of nonvertebral fractures [68].

However, the administration of bisphosphonates orally possesses some common side effects like dysphagia, nausea, abdominal pain, constipation or diarrhea, flatulence, acid regurgitation, esophageal ulcers, taste distortion, and gastritis [69,70]. Patients are advised to take the medication on an empty stomach, usually in the morning after an overnight fast, and remain in an upright position for at least 30 min of oral administration to avoid esophageal irritation. Moreover, poor drug bioavailability is another issue related to oral administration due to high drug hydrophilicity [71]. To alleviate gastrointestinal disturbances, the number of oral doses might be reduced, or the administration route might be changed to pre-prandial administration or intravenous administration [70]. Rarely, bisphosphonates might cause osteonecrosis of the jaw (ONJ), atypical femoral fractures (AFFs), influenza-like symptoms, uveitis, hypocalcemia, and episcleritis [69,70]. To alleviate the rare side effects, the dose might be decreased, or the schedule of the dosing might be changed, as bisphosphonates have a lasting effect after treatment discontinuation [70,72]. As bisphosphonates suppress osteoclast activity, that leads to a reduction in turnover rate and remodeling rate, which might increase micro-damage accumulation in the bone and atypical fracture susceptibility in long bones in long-term administration [73].

In conclusion, bisphosphonate is a successful therapeutic choice for those with early osteoporosis despite some side effects as long as the administration strategy is strictly followed [16,74]. Nonetheless, the long-term effects and risks of bisphosphonates beyond 5 years remain unknown [73], and further study is required to establish the appropriate time length for bisphosphonate administration.

#### 3.1.2. Estrogen Replacement Therapy/Hormone Replacement Therapy (ERT/HRT)

Endogenous estrogen contributes to bone mass maintenance, and its absence after menopause has been linked to fast bone loss [75]. Strong evidence suggests that estrogen, the anti-resorptive drug, acts through estrogen receptor α (ERα) to induce osteoclast apoptosis and decrease apoptosis in osteoblasts and osteocytes [75]. Estrogen binds to ERα, and then it transfers to the osteoclast nucleus, contacts the Fas ligand site of transcription, and promotes the transcription of FasL, which contacts the Fas receptor on the pre-osteoclast surface, leading to caspase 8 cleavage, which stimulates osteoclast apoptosis [76]. When estrogen binds to the ER receptor, it inhibits RANKL and boosts the synthesis of OPG [37]. At various phases of the osteoblast lineage’s differentiation, the deletion of ERα in osteoblasts showed varying effects [77]. Consequently, the deletion of ERα in osteoblast progenitors or from mesenchymal progenitors leads to an inhibition of periosteal osteoblast progenitor cell proliferation and differentiation [77]. The estrogen stimulation effect on progenitor cell proliferation and differentiation is due to potentiating Wnt/β-catenin signaling [78]. However, ERα deletion in osteocytes or osteoblasts does not have any effect on bone architecture or bone mass [78]. Interestingly, HRT is comparable to ERT, but it includes progestin along with estrogen instead [76]. However, women who need hormone treatment after a hysterectomy are advised to use estrogen alone, but progestin–estrogen combination therapy is recommended for those who still have an intact uterus to avoid the negative effects on the endometrium [76]. Healthy postmenopausal women with normal bone density who had previously taken part in placebo-controlled HRT studies were analyzed, and the finding was that the HRT-treated women had a considerably lower risk of osteoporotic fractures when compared to the placebo-treated women [79]. Indeed, vertebral and hip fractures decreased by 34% and other fractures by 23% after HRT with medroxyprogesterone (2.5 mg) and equine estrogen (0.625 mg) for five years [68].

However, both HRT and ERT have negative side effects with long-term usage. ERT increases the chance of stroke, endometrial cancer, and venous thromboembolic disorders. HRT increases the risk of cerebrovascular accidents, cardiovascular diseases, breast cancer, and venous thromboembolic disorders [72]. Tibolone, a new regimen of HRT, consists of a steroid that is degraded by the biological metabolism process into estrogen, progesterone, and testosterone [80]. Tibolone alleviates menopausal symptoms like bone density loss, vaginal atrophy, osteoporosis, and hot flushes; however, it increases the risk of cerebrovascular accidents with long-term use [80]. To overcome these side effects, researchers introduced selective estrogen receptor modulators (SERMs) [12]. Moreover, breast cancer and cerebrovascular accident risks are still a problem with the long-term usage of SERMs [81].

In conclusion, only when the benefits exceed the hazards should postmenopausal women with osteoporosis consider using ERT/HRT [16]. The use of ERT/HRT for an extended period should be avoided. The patient’s therapy should be changed to another medication when the ERT/HRT is discontinued [82].

#### 3.1.3. Selective Estrogen Receptor Modulators (SERMs)

SERMs are introduced to avoid the adverse effects of long-term estrogen treatment. SERMs are nonsteroidal medicines that attach to estrogen receptors and display selective estrogenic action based on cell or tissue type [38]. SERMs have the same mechanism of action as estrogen therapy, as they enhance osteoclast apoptosis without side effects on the endometrium and breast [38]. The approved SERMs by the FDA are raloxifene, which is used to treat and prevent osteoporosis, and bazedoxifene, which is used to prevent osteoporosis only in combination with estrogen [12]. As these medications decrease vertebral fracture risk, however, they cannot effectively treat hip or nonvertebral fractures [12].

Interestingly, SERMs can positively affect the skeletal system in males, too [83]. One of the main causes of male osteoporosis in adult men is hypogonadism, or a lack of testosterone [84]. When testosterone binds to the androgen receptor (AR), it directly affects bone metabolism. However, it can also have an indirect effect by aromatizing estradiol to activate the estrogen receptor-α and or -β (ER-α, ER-β) [85]. Compared to testosterone, estrogen has been demonstrated to have a more significant and independent impact on bone mineral density [86]. Significantly, SERMs improve BMD, bone histomorphometric indices, BMC, and microstructures [83].

However, SERMs raise the risk of thromboembolic disorder, muscular spasms, and stroke [87]. Additionally, when SERM treatment is discontinued, the remodeling rate will quickly rise, leading to quick bone loss and an increased risk of fracture [88]. Also, due to the observed adverse effects, the use of SERMs to treat male osteoporosis should be approached with caution [83].

In conclusion, SERMs are considered the first-line treatment for osteoporosis for postmenopausal women [76]. However, they are recommended only for relatively young women with high vertebral fracture risk and low risk of nonvertebral fracture and deep vein thrombosis [82]. Similar to estrogen treatment, long-term use of SERMs is avoided, and a substitute drug is needed after SERM discontinuation [82].

#### 3.1.4. Calcitonin

Calcitonin is a peptide hormone released by thyroid C cells [18]. Calcitonin decreases calcium reuptake in the small intestine and kidney and increases calcium absorption into the bone, ending with more calcium stored in bones [18]. Calcium combines with phosphoric acid to make hydroxyapatite, which gives the strength and hardness of the bone [18]. Calcitonin stimulates Wnt10b in osteoclasts, leading to bone formation induction [89]. Moreover, calcitonin binds to osteoclast receptors, limiting osteoclast motility and capacity to resorb bone via transcription regulation by (cAMP)/(PKA)-cAMP-response element binding protein pathway [18,90]. It also inhibits the maturation of osteoclast precursors, leading to bone loss [90]. Additionally, calcitonin has a pain-relieving action by changing the serotonergic systems, normalizing the sodium channel, and decreasing the disturbance of the peripheral circulatory [91]. Interestingly, calcitonin’s activities are opposed to those of PTH; PTH increases blood calcium while calcitonin reduces it [92].

However, calcitonin might cause hypocalcemia, loss of appetite, abdominal pain, nausea, and diarrhea [38]. Calcitonin causes hypocalcemia as a result of its decreasing effect on calcium reuptake of the small intestine and kidney and its increasing effect on calcium absorption into the bone [38]. Consequently, it is necessary to carefully monitor the blood calcium level, and the administration of vitamin D along with calcitonin may prevent hypocalcemia [38]. Moreover, calcitonin also exhibits an increasing incidence of prostate cancer [38]. Even though a weak correlation exists between prolonged calcitonin usage and malignant tumors [93], clinicians’ decisions will probably be influenced by this weak potential.

In conclusion, calcitonin medication is typically administered as a second-line treatment, and it is used as a short-term treatment [76]. Alternative long-term treatments, such as bisphosphonate, must be used in place of calcitonin therapy if side effects occur [76].

#### 3.1.5. Anti-RANKL Antibody

The identification of the receptor activator of the NF-B ligand (RANKL) pathway as a key regulator of bone resorption has contributed significantly to the knowledge of the molecular control of bone physiology [94]. RANKL, a soluble glycoprotein that is a member of the TNF family, is expressed via osteocytes and osteoblasts in bone tissue [94]. Consequently, the trimeric RANKL binds to the RANK receptor, causing stimulation of osteoclast differentiation, survival, and involvement in bone-resorbing activity [95]. OPG competes with RANK to regulate bone resorption. RANKL has quickly gained interest as a target for treating osteoporosis due to its regulatory function in bone resorption [95]. Denosumab, a human monoclonal antibody, is the first and only RANKL antagonist to be authorized by the FDA, as well as the first antibody treatment licensed for postmenopausal osteoporosis therapy [96]. Later, denosumab was licensed for the management of osteoporosis in males, bone loss caused by aromatase antagonists, and glucocorticoid-induced osteoporosis [97]. Denosumab suppresses RANK-mediated osteoclastogenesis and hence increases bone mass by blocking RANKL binding to RANK receptors [96]. To prove the effectiveness of denosumab, research named “Fracture Reduction Evaluation of Denosumab in Osteoporosis every 6 Months (FREEDOM)” was performed, which indicated an elevation in BMD and a reduction in bone resorption rate in the denosumab group versus the placebo group [98]. Other studies reported a reduction in fracture risk and an increase in BMD at the femoral neck, lumbar spine, total hip, and distal radius after 24 months of denosumab prescription [99,100,101]. Despite denosumab’s efficiency in decreasing the risk of fracture and increasing BMD in long-term management, it is not recommended as a first-line therapy for osteoporosis because denosumab weakens the immune system. Denosumab affects lymphocytes that need RANK/RANKL connections by targeting such interactions, which lowers lymphocyte activity and raises the probability of infection [76,102]. Denosumab is frequently used instead of bisphosphonates for individuals with renal insufficiency due to bisphosphonates’ side effects on the kidneys [76,102].

However, denosumab’s beneficial skeletal effects might be reversed rapidly after discontinuation [103] because of a rebound enhancement in osteoclastogenesis [104], resulting in a significant increase in bone turnover, which is usually above pre-treatment rates in a process known as the “rebound phenomenon” [105]. In addition, the FREEDOM trial also emphasized the long-term denosumab side effects, which were comparable to those encountered with bisphosphonates, such as atypical femoral fractures (AFFs), osteonecrosis of the jaw (ONJ), musculoskeletal discomfort, hypocalcemia, and gastrointestinal problems [98]. Unlike bisphosphonates, denosumab impairs the immune system [38]. Denosumab affects lymphocytes that need RANK/RANKL connections, resulting in reduced lymphocyte function and an increased infection risk [38].

In conclusion, subsequent anti-resorptive therapy is required after denosumab discontinuation to avoid the rebound phenomenon. Consequently, it is advised that subsequent bisphosphonate therapy lasts between one and two years [105]. The entire length of denosumab administration may influence how well the subsequent bisphosphonate therapy inhibits bone loss [105]. Moreover, patients who have vertebral fractures are more vulnerable to bone loss and rebound-related vertebral fractures after stopping denosumab [105].

#### 3.1.6. Cathepsin K Inhibitors

Cathepsin K, a papain-like cysteine protease family member, is a novel target for the development of anti-resorptive therapies [106]. As a human protease, cathepsin K is produced in osteoclasts during bone remodeling to decompose type I collagen synthesized in bone matrix and promote resorption [106]. One of the cathepsin K inhibitors is odanacatib. Patients treated with odanacatib had a steady rise in BMD at several locations up until year 5, after which the BMD was stabilized or slightly enhanced. In comparison to the baseline, the rate of bone turnover also declines [107]. Moreover, odanacatib has shown equivalent efficacy to bisphosphonates with little impact on bone formation [108]. Another cathepsin K inhibitor is balicatib, which has the ability to enlarge osteoclasts and reduce collagen degradation [109]. Unlike other anti-resorptive agents, cathepsin K inhibitors decrease osteoclast activity without a reduction in osteoclast number [110].

However, the full side effects related to cathepsin K inhibitors are still under investigation, but AFF, pycnodysostosis, and a higher risk of cerebrovascular accident are among the reported negative effects. Further research on the side effects is required [38].

In conclusion, cathepsin K inhibitors are efficient anti-resorptive drugs, but more studies are needed regarding their side effects [38].

### 3.2. Anabolic Drugs

Anti-resorptive therapies cannot directly increase bone formation due to their mechanism of action, which is based on the suppression of osteoclast activity and the concomitant decrease in bone remodeling. As a result, when treating individuals with substantially deteriorated bone quality, anti-resorptive therapies will not be the first option [111]. Consequently, anabolic therapies that enhance bone formation may hold the answer to solving this problem.

#### 3.2.1. Parathyroid Hormone PTH

PTH, an anabolic drug, is a hormone released by the parathyroid glands to boost serum calcium levels. PTH increases serum calcium levels by promoting bone resorption and increasing osteoclast activity, ending with the release of calcium from bones into the blood [47]. PTH appears to exacerbate the condition of people with osteoporosis at first look. However, researchers discovered that its function of increasing bone remodeling occurs only when its receptor is constantly active [112]. PTH1 receptor (PTH1R), a G protein-coupled receptor (GPCR) found in osteoblasts, regulates PTH function [112]. As PTH1R is activated, it stimulates multiple GPCR-related signaling, including the PLC/PKC, cAMP/PKA, and ERK, leading to increased bone remodeling [112]. PTH also affects the Wnt signaling pathway by inhibiting sclerostin, a Wnt antagonist [112]. PTH effects are classified as either anabolic or catabolic. PTH’s anabolic effect promotes osteoblast proliferation and differentiation, resulting in increased bone production, but the catabolic effect indirectly promotes bone resorption because osteoclasts are triggered by RANKL released by osteoblasts [18,113]. The time when bone formation is greater than bone resorption is known as the anabolic window, during which maximal bone formation takes place. When a PTH is delivered, bone formation markers are first raised, and bone resorption markers are activated subsequently. Bone resorption gradually accelerates just after the anabolic window. As a result, the PTH receptor can promote bone formation when activated intermittently and at low doses [114]. This two-sided action is mostly owing to the PTH receptor’s two distinct conformations: R0 for extended activation, which causes osteoclast activation, and RG for intermittent activation, which increases bone remodeling [115]. The FDA authorized two PTH analogs: teriparatide, a full-length PTH fragment, in 2002, and abaloparatide, a PTH-related peptide (PTHrP), in 2017 [116]. Abaloparatide appears to be more powerful than teriparatide, possibly because of its increased affinity for RG [117].

Teriparatide (PTH1-34), which is called a PTH analog, is composed of the first 34 N-terminal amino acids of PTH. It is well known that administering PTH1-34 continuously has a catabolic impact, whereas doing so intermittently has an anabolic effect on bone [118]. The FDA has certified teriparatide to treat postmenopausal osteoporotic women and osteoporotic men who are at high fracture risk. Teriparatide is also licensed to treat osteoporosis associated with long-term systemic glucocorticoid treatment in both men and women who are at high risk of fracture. Men with primary or hypogonadal osteoporosis who are at high risk of fracture can also be treated with teriparatide to improve bone mass [68]. Additionally, abaloparatide (PTHrP1-34), which is called a PTHrP analog, is a PTH-related protein analog that shares similarities with PTH1-34 as both have comparable secondary structures and share eight of the first 13 amino acids [117]. Both PTH receptor 1 and the PTHrP-specific receptor are involved in PTHrP1-34 actions [117].

However, long-term excessive doses of teriparatide increase the osteosarcoma risk [119]. PTH analogs might cause cephalgia, dizziness, limb cramps, and nausea [119]. Abaloparatide has side effects similar to teriparatide side effects, including injection-site reactions, gastrointestinal complaints, myalgia, and dizziness [120]. The excessive use of abaloparatide is associated with osteoblastoma and osteosarcoma [120]. However, teriparatide induces hypercalcemia more than abaloparatide [116].

In conclusion, despite the PTH effect against nonvertebral and vertebral fractures, the side effects of parathyroid hormonal therapy, particularly the increase in osteoclast activity in long-term use, restrict the use of PTH to no longer than two years to treat osteoporosis [121]. It is crucial to switch to different medications as significant bone loss is being noticed during the period of discontinuation [121]. Due to their cost and difficulties in administering subcutaneous injections, it is not advised as a first-line treatment for osteoporosis [119].

#### 3.2.2. Anti-Sclerostin Antibody

Sclerostin is an osteocyte-secreted glycoprotein that inhibits the canonical Wnt signaling pathway and acts as an antagonist of bone morphogenetic protein-7 (BMP7) [122]. Consequently, sclerostin is removed from the Wnt signaling pathway by anti-sclerostin antibody therapy, which activates the canonical Wnt signaling pathway, leading to increasing bone formation and reducing bone resorption [112]. Sclerostin also combines with proBMP7 and mature BMP7 to promote intracellular accumulation and degradation of BMP7 [122]. Sclerostin suppresses the BMP7 signaling, which is recognized for its capacity to promote the formation of bone and cartilage [122]. As a result, anti-sclerostin therapy inhibits sclerostin from BMP7 suppressing, leading to bone formation stimulation. Recently, sclerostin blocking increases DKK-1, which competitively binds with LRP5/6 and inhibits Wnt signaling and bone formation [123], as a negative response, leading to a reduction in the anabolic effect of anti-sclerostin [124]. Therefore, even in individuals with osteoporosis, the use of bispecific antibodies against both DKK-1 and sclerostin is anticipated to stimulate bone formation more than treatment with just an anti-sclerostin antibody [124]. However, there is no clinical research using bispecific antibodies and DKK-1 antagonists to treat osteoporosis yet [124]. Osteoporosis has been treated with several anti-sclerostin antibodies, such as romosozumab. Romosozumab, a sclerostin monoclonal antibody, is an FDA-approved treatment for postmenopausal osteoporotic women at a considerable risk of fracture [125]. Romosozumab increases BMD and decreases the risk of fracture. Indeed, romosozumab, followed by alendronate therapy, decreases hip fracture more than alendronate therapy alone [125]. Moreover, romosozumab, unlike PTH analogs, has the potential to both increase bone formation by regulating the Wnt pathway and decrease bone resorption because sclerostin also increases the production of RANKL [126]. Furthermore, unlike other osteoporosis treatments, the main romosozumab effect is modeling, not remodeling-based, providing it an extraordinary potency when it is used alone or combined with other treatments [125,126].

However, the most frequent side effects of anti-sclerostin antibody therapy include myocardial infarction, stroke, and cardiovascular events [127]. The (FRAME) study, a Fracture Study in Postmenopausal Women with Osteoporosis, also revealed that Wnt signaling is linked to malignant tumors [122,124,126,127].

In conclusion, due to the serious side effects of anti-sclerostin antibodies, long-term treatment is not recommended since it puts the heart in danger and might result in malignant tumors [112,127].

#### 3.2.3. Strontium Ranelate

Calcium-sensing receptors (CaSRs) are located in parathyroid glands, osteoblasts, and osteoclasts [128]. Blocking of CaSRs in the parathyroid gland leads to stimulation of PTH secretion and activation of PTH receptors on bone cells, resulting in an increase in BMD [128]. However, blocking CaSRs in the parathyroid gland has a weak increasing effect in BMD compared with other treatments, such as alendronate; hence, CaSRs in the parathyroid gland are no longer a valid therapeutical target [129]. Moreover, activation of CaSRs in osteoblasts has a potential stimulation effect on bone formation. CaSRs in osteoblasts and osteoclasts are controlled via extracellular calcium concentration [130]. A high level of extracellular calcium stimulates various signaling pathways, including protein kinase C (PKC), phospholipase C (PLC), JNK, ERK, PKA, and cAMP [131]. Consequently, ERK signaling improves osteoblast proliferation, and AKT signaling reduces osteoblast apoptosis [131]. Additionally, high extracellular calcium concentration stimulates the secretion of insulin-like growth factors (IGF-1 and IGF-2), causing stimulation of osteoblast proliferation and differentiation by boosting the expression of prostaglandin E2 and cyclooxygenase 2 [132]. Moreover, stimulation of CaSRs in osteoclasts activates PLC and NF-κB, causing osteoclast apoptosis [133]. Consequently, CaSR activation leads to an anabolic and anti-resorptive effect on the bone cells. Significantly, strontium ranelate is used as a second-line treatment for osteoporosis as it is considered a CaSR activator in osteoblasts and osteoclasts [134]. Strontium has a nucleus size the same as that of calcium, so it is absorbed by cells instead of calcium and transported to the bone [135]. Therefore, strontium ranelate enters osteoblasts and osteoclasts via CaSRs and acts identically to calcium in its action [135], causing an increase in osteoblast proliferation and differentiation. Also, it activates OPG production through osteoblasts, reducing osteoclast activity [135]. Furthermore, strontium ranelate can also bind directly to CaSRs of osteoclasts and osteoblasts, increasing osteoclast apoptosis and osteoblast proliferation and differentiation [135].

Strontium ranelate, known for causing severe side effects such as venous thromboembolism, cardiovascular disorders, nervous system symptoms, and myocardial infarction, was discontinued by its manufacturer in May 2017 [136]. This decision followed numerous health warnings over the years, including alerts about potentially fatal allergic reactions, increased risks of cardiac issues, and venous thromboembolism. Despite its discontinuation, recent EPACT data indicate that several GP practices in East Lancashire were still prescribing this medication. Therefore, healthcare professionals must be aware that strontium ranelate is no longer available. Patients currently taking this medication should be evaluated to plan for suitable alternatives.

#### 3.2.4. Pro-Inflammatory Cytokines Complex Network Inhibitors

Tumor Necrosis Factor-alpha (TNF-α) Inhibitors:

As the first medication was used to treat rheumatoid arthritis, TNF-α inhibitors are the most often utilized in clinical practice. Interestingly, TNF-α blockage causes inhibition in generalized bone loss in inflammation-induced osteoporosis [137]. Infliximab, a TNF-α antagonist, has shown paradoxical results in several clinical studies. The bone formation marker (osteocalcin) and bone resorption marker [N-terminal telopeptide (NTX)] were inhibited at week 14 as compared to the baseline levels. The deoxypyridinoline (DPD) (bone resorption marker) was also inhibited after six months of infliximab prescription. There was no significant clinical modification in bone marker levels between the 6th and the 12th month, showing that infliximab had short-term effects [138]. In contrast, another clinical study reported that infliximab stimulated N-propeptide of type I procollagen (P1NP) and osteocalcin, which are bone formation markers, and inhibited C-terminal cross-linking telopeptide of type I collagen (ICTP), which is a bone resorption marker in the 6th week of the treatment [139]. Another study has claimed that treatment with infliximab for one year causes a significant decrease in CTX-1 level and a significant increase in osteocalcin level [140]. Clinically, infliximab treatment for one year leads to a non-significant change in BMD of the hip and lumbar spine [139]. However, these variations in the studies’ outcomes might return to the study population. The first study examined women with a mean age of 48 years who might be in the menopausal period, so bone markers might vary during the menopausal period. Generally, infliximab therapy leads to serum level changes for bone resorption and formation markers, leading to improved bone remodeling.

Interleukin-6 Inhibitors:

Interleukin-6 inhibitors are approved treatments for rheumatoid arthritis and have an inhibition effect on osteoclast activity in inflammation-induced osteoporosis, leading to bone mass influence. Tocilizumab, an IL-6 inhibitor, showed a stimulation effect on PINP (a bone formation marker) after four weeks of treatment. Furthermore, it exhibited an inhibitory effect on CTX-1 and ICTP (bone resorption marker) at the 4th, 16th, and 24th weeks of the treatment [141]. Another large clinical study has illustrated that tocilizumab had a reducing effect on the CTX-1/osteocalcin ratio in the 24th week [142]. Moreover, a long-term clinical trial has found that tocilizumab treatment for 2 years increases BMD of the femoral neck, but the treatment does not cause any change in BMD at the lumbar spine. Tocilizumab treatment also caused CTX to decrease with no change in osteocalcin or P1NP [143]. However, another clinical study has reported there was a significant increase in BMD in both the lumbar spine and the femoral neck in patients who have osteopenia but no change in BMD at either femoral neck or lumbar spine for patients who do not suffer from osteopenia [144]. Generally, tocilizumab can reduce bone resorption markers in inflammation-induced osteoporosis and improve bone mass when the treatment continues for 2 years. However, there is not adequate information on the effect of kevzara, another IL-6 inhibitor, on bone mass and bone markers [145].

Interleukin 17 Inhibitor:

Secukinumab is a monoclonal antibody that selectively interacts with and neutralizes IL-17A, which is a member of the IL-17 family [145]. According to a clinical study, secukinumab medication had little effect on bone turnover markers (BTM) during the first 6 months of treatment, but it had a significant impact on fine bone cell activity moderators like WNT inhibitors and was able to somewhat restore Dkk-1 levels in psoriatic arthritis patients. However, additional investigations involving a larger sample size are required to evaluate whether these early findings have clinical value. Also, secukinumab treatment causes a reduction in the bone structural change in inflamed bone [145]. Consequently, further studies are required to study the effect of secukinumab on bone mass and bone markers.

However, congestive heart failure and aplastic anemia are two negative side effects of using both TNF-α and IL-6 inhibitors [146,147]. The two therapies possess low tissue penetrating capacity and oral availability [148].

In conclusion, more studies are requested to clearly illustrate the effectiveness of the pro-inflammatory cytokines inhibitors in treating osteoporosis.

### 3.3. Nutritional Supplements

#### 3.3.1. Calcium

Although calcium is an essential mineral for healthy bones, the diet often lacks calcium [149]. A low amount of serum calcium will increase PTH production, eventually resulting in a high bone turnover rate [149]. Calcium administration, on the other hand, will lower PTH secretion and finally hinder bone resorption [149]. Moreover, calcium supplementation is best reserved just for individuals with secondary hyperparathyroidism or whose osteoporosis pathogenesis is clearly correlated to calcium deficiency [149]. Also, calcium treatment for osteoporosis is typically used in addition to anti-RANKL or bisphosphonate medications [149]. However, calcium is a threshold mineral, meaning that extra ions are eliminated. Therefore, excessive calcium consumption has no positive effects on bone health. Additionally, a meta-analysis illustrated a 15% decrease in the overall osteoporotic fractures’ summary relative risk estimations when a combination of calcium and vitamin D is used [150]. A cross-sectional study has reported that there is a positive association between calcium supplements and bone health (*p* < 0.05) [151]. Moreover, taking calcium supplements might seem to speed up the healing of osteoporotic bone fractures, but they do not increase bone strength [152].

However, the common side effect of calcium use is gastrointestinal disorders [149]. The combination of calcium and vitamin D might cause hypercalcemia; thus, serum calcium level monitoring is important during the usage of the combination of calcium and vitamin D [149]. The combination of calcium and vitamin D might also lead to urinary stone incidence, but that incidence is not associated with the use of calcium alone [149]. Therefore, the effect of calcium supplementation in preventing osteoporotic fractures is limited even though they have a minimal risk of serious and moderate adverse effects [153].

#### 3.3.2. Vitamin D

Vitamin D promotes the intestinal intake of calcium, phosphate, and magnesium, which is crucial for building and maintaining bone strength [154]. The two major forms of vitamin D are D2, known as ergocalciferol, which is derived from plants, and D3, known as cholecalciferol, which is generated in the skin from sunlight and ingested from animal meals [155]. Moreover, vitamin D3 is considered the most crucial type of vitamin D, and the end-product of vitamin D3 ingestion is the active form of calcitriol that binds to vitamin D receptor (VDR) in the guts, kidneys, parathyroid gland, and bones, causing increasing serum calcium levels. The risk of vitamin D shortage increases with age as the ability of vitamin D production through the skin is decreased [155]. Additionally, vitamin D deficiency might cause muscle fiber atrophy of type II, increasing the potential for fractures as well as the tendency to fall [155].

Combined therapy involving calcium and vitamin D can lead to several side effects, including hypercalcemia, gastrointestinal issues, renal calculus, and myocardial infarction, which often outweigh the therapy’s limited benefits [156]. Conversely, monotherapy with vitamin D does not exhibit any non-skeletal side effects [153].

Vitamin D3 deficiency is linked to low bone mineral density (BMD) and defects in bone mineralization, which increase the risk of fractures [157]. Additionally, vitamin D3 deficiency can cause non-skeletal effects such as acute respiratory infections, muscle weakness, and falls [157]. Consequently, The European Calcified Tissue Society recommends improving vitamin D status by incorporating it into bread, dairy products, and cereals, particularly for vulnerable groups, including infants, children up to three years old, older adults, pregnant women, and non-Western immigrants [157].

#### 3.3.3. Vitamin K2

There are two kinds of vitamin K, including vitamin K1 (phylloquinone) and K2 (menaquinone). Interestingly, vitamin K2 is primarily generated in the human body from vitamin K1. Consequently, vitamin K1 insufficiency leads to a vitamin K2 shortage [158]. Moreover, vitamin K2 normally contains unsaturated isoprenyl side chains, which are different in length and are known as menaquinone-n. Significantly, menaquinone-4 (menatetrenone) has been studied the most [40]. Menaquinone-4 is considered the active form of vitamin K2, and it aids in the γ-carboxylation of osteocalcin, which is released via osteoblasts to form a bone matrix [40]. Independently of BMD, vitamin K shortage raises the chance of fracture in older women. Vitamin K deficiency also leads to an increase in the chance of fracture during bisphosphonate therapy [159]. A modest enhancement of lumbar spine BMD is reported during monotherapy of menatetrenone [160]. Fortunately, vitamin K2 is thought to be safe with few adverse effects [158]. In conclusion, both vertebral and nonvertebral fractures can be avoided with menatetrenone without side effects [158].

#### 3.3.4. Vitamin E

Vitamin E encompasses a group of fat-soluble compounds, including tocopherols and tocotrienols, each with distinct antioxidant properties and potential implications for bone health, including osteoporosis [161]. Tocopherols, consisting of alpha, beta, gamma, and delta forms, are the most studied and abundant in the diet. Tocotrienols, structurally similar to tocopherols but possessing an unsaturated side chain, are less prevalent in the diet but exhibit potent antioxidant and anti-inflammatory effects [161]. Vitamin E may be able to eliminate free radicals before they trigger NFκB, inhibiting the generation of cytokines and, eventually, osteoporosis. Vitamin E has also been demonstrated to suppress COX-2, an enzyme that has a role in inflammatory reactions [162]. Moreover, metabolic syndrome primarily promotes sRANKL, SOST, DKK-1, and FGF-23 levels, resulting in an imbalance in the bone remodeling process and bone loss. Consequently, an in vivo study has shown that 12 weeks of treatment with tocotrienol can correct the alterations in these bone-related peptides produced by metabolic syndrome, confirming the tocotrienol’s anti-osteoporotic potential [163]. Tocotrienol does not have any remarkable side effects [164].

### 3.4. Stem Cells

The growing interest in utilizing stem cell therapy for various diseases is largely due to their self-renewal capabilities, which can potentially regenerate and heal damaged tissues [165]. Among these, stem cells are considered a primary source of bone cell replacement therapies in osteoporosis. They are broadly categorized into three types: induced pluripotent stem (iPS) cells, embryonic stem (ES) cells, and somatic stem cells such as mesenchymal stem cells (MSCs). MSCs, in particular, have garnered significant attention in osteoporosis treatment due to their ease of isolation and fewer ethical and safety concerns compared to ES cells and iPS cells [165]. The therapeutic action of MSCs in bone formation following transplantation is believed to involve two main processes: (1) MSCs directly adhering to the affected bone areas and differentiating into osteoblasts; and (2) MSCs releasing various growth factors that inhibit osteoclastic differentiation and promote angiogenesis, thereby indirectly contributing to the repair of damaged sites [166].

Despite these advantages, there are currently no well-documented side effects associated with using MSCs to treat osteoporosis, and they are approved by the FDA. However, the use of iPS cells is linked to concerns about tumorigenesis. Thus, the application of iPS cells in treating osteoporosis requires further investigation to fully understand potential risks and develop appropriate safety measures. More research is essential to explore the side effects and confirm the long-term safety of MSC-based therapies [167].

### 3.5. Combination Therapy

Numerous investigations have been carried out to identify more efficient treatments since pharmaceutical medications for osteoporosis have limits [168]. A suitable example of this is combination treatment, which combines either two anti-resorptive medications or an anti-resorptive medication with an anabolic medication in hopes of having a synergistic effect [168]. Several trials used available medications to investigate combination treatment [168]. A combination of PTH and alendronate showed no improvement in BMD when compared to the usage of either drug alone [168]. Another trial that included PTH and SERMs likewise found no improvement in BMD [169]. Contrarily, BMD was slightly elevated when denosumab and teriparatide were combined [87,170]. Teriparatide can be administered alone or in combination with denosumab and abaloparatide, according to different research [87]. Previous research has shown that PTH and PTHrP analogs have a significant role in the management of osteoporosis, whether used alone, in combination, or in sequence with anti-resorptive medicines [171].

However, when compared to monotherapies, there is a worry that combination medicines may raise the chance of serious side effects [172].

In conclusion, combination treatment is typically not advised for osteoporosis because of its combined side effects and higher expense, as the benefit is just a small rise in BMD [171]. Combination treatment is thus only appropriate for individuals at high risk of fracture or when other treatments have failed [171].

### 3.6. Sequential Therapy

As previously mentioned, extended usage of monotherapy for osteoporosis is not recommended since it is linked to several issues, including serious adverse effects and the loss of anti-osteoporosis efficacy [82]. Sequential therapy is, therefore, the first option to explore to avoid the discontinuation effects [82]. After discontinuing denosumab or estrogen/SERM, bisphosphonates have previously been utilized in clinical practice to stop the fast increase in the bone remodeling rate [82]. The sequence is crucial when switching from anabolic to anti-resorptive medications. In clinical research, the femoral neck and total hip BMD in patients treated with teriparatide followed by denosumab or a combination of both drugs at the same time were higher than in patients treated with denosumab followed by teriparatide even though the lumbar spine BMD was same [173].

However, sequential therapy has side effects, the majority of which seem to be comparable to those of monotherapy [82].

In conclusion, sequential therapy is beneficial against osteoporosis because it enhances BMD in comparison to the effects of continuous monotherapy.

### 3.7. Alternative Therapy

Some studies have approved some adjuvant treatments that might help to improve the quality of life of patients who have osteoporosis. In a cross-sectional study of 50 osteoporotic patients, the participants who performed regular physical activities for at least three days per week improved their muscle strength, leading to reaching an ideal body mass index (BMI), preserving the lumbar (BMD) and preventing fractures in osteoporotic postmenopausal women [174]. Another cross-sectional study has shown that there is a negative association between exercise barriers and bone health [151]. A single-blinded pilot randomized controlled study was conducted for eight weeks on 30 postmenopausal women who were suffering from low bone mass. This study has illustrated that an efficient way to increase back extensor muscle (BEM) strength and physical performance in postmenopausal women is through physiotherapeutic education, which combines group instruction with therapeutic home exercises [175]. Overall, patients who carry out regular physical exercises are less suspectable to bone fractures and have better bone health.

Significantly, some medicinal plants have shown a high potential to treat and/or prevent osteoporosis. Finding natural remedies that may have bone-protective properties could provide therapy options that would address the limitations of conventional medicines [176]. In recognition of its estrogenic effects, especially its involvement in the treatment of osteoporosis, labisia pumil (LP), also known as Kacip Fatimah in Malaysia, has drawn much interest [177]. Demethylbelamcandaquinone B, the bioactive isolated compound from LP, can replicate the mouse osteoblastic cell line (MC3T3-E1) by accelerating the proliferation and differentiation process in an in vitro study [177]. Indeed, kaempferol, the natural flavonoid derivative from medicinal plants such as Moringa oleifera and Ginkgo biloba, has anti-inflammatory, anti-oxidative, and anti-osteoporotic effects [178]. It has been suggested that some of the fundamental mechanisms responsible for kaempferol’s anti-osteoporotic effects include (a) a reduction in adipogenesis, which supports osteoblastogenesis and chondrogenesis, (b) the stimulation of the estrogen receptor signaling pathway, (c) a boost in Runx-2 expression, which serves as the principal osteogenic transcription factor in BMP-2 signaling, (d) the suppression of the inflammatory reaction through reduction in NF-κB, (e) the decrease in lipid peroxidation and intracellular ROS generation, (f) the differential regulating of osteoclast and osteoblast autophagy, (g) the inhibition of osteoblast apoptosis, and (h) the modulation of proteins related to osteoblast mineralization. All of these molecular activities contribute significantly to the preservation of tightly connected bone formation and resorption [179].

## 4. Conclusions

In conclusion, this comprehensive review has shed light on the mechanisms underlying the efficacy and limitations of existing medications for osteoporosis. Figure 10 illustrates the integration between the crucial pathways and pharmacological therapies of osteoporosis while Table 1 summarizes their mechanisms of action along with the side effects. Although bisphosphonates remain the first-line treatment due to their favorable safety profile, the lack of long-term data beyond five years poses a limitation. Estrogen replacement therapy/hormone replacement therapy (ERT/HRT) presents an alternative but is associated with significant risks, such as cerebrovascular accidents and malignant tumors, leading to the exploration of selective estrogen receptor modulators (SERMs) as safer alternatives. However, SERMs may lead to rapid bone loss upon discontinuation, necessitating further investigation into their long-term effects. Denosumab, an anti-RANKL antibody, offers promise for patients with renal dysfunction who cannot tolerate bisphosphonates, while odanacatib shows comparable efficacy to bisphosphonates with minimal impact on bone formation. Nonetheless, the potential side effects of cathepsin K inhibitors warrant further study. Combination therapies and sequential treatment strategies are employed to optimize efficacy while minimizing side effects, yet more research is needed in these areas. Furthermore, emerging modalities such as stem cell therapy hold potential but require further investigation regarding safety and efficacy. Finally, attention should be directed toward exploring the anti-osteoporotic effects of medicinal plants to develop novel therapies with fewer side effects, ultimately benefiting both patients and healthcare providers. Continued research efforts aimed at improving osteoporosis management are essential for advancing patient care in the future.

## Figures and Tables

**Figure 1 biomedicines-12-01635-f001:**
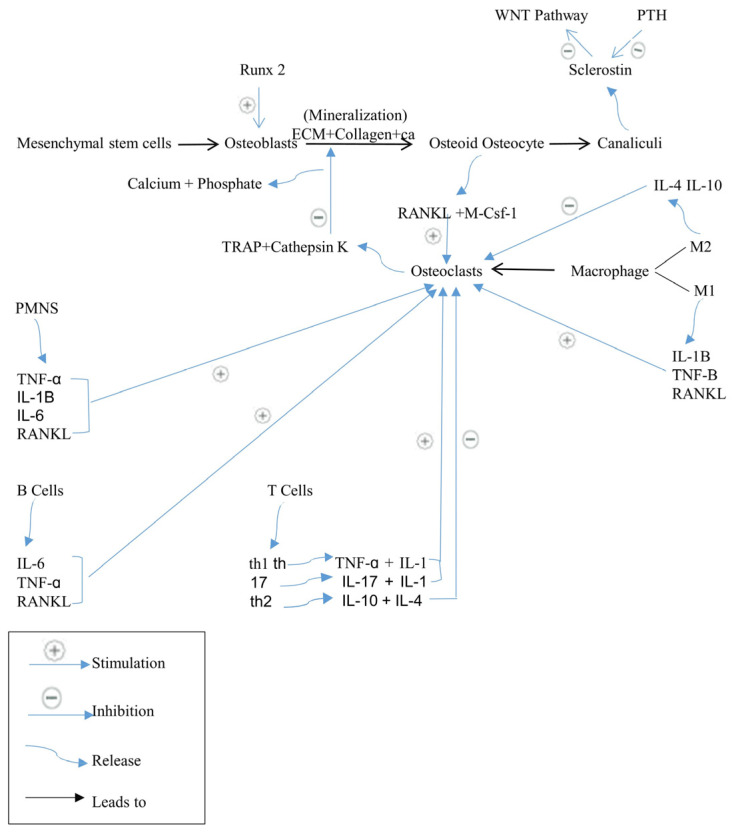
The cellular mechanisms responsible for the control of bone remodeling process.

**Figure 2 biomedicines-12-01635-f002:**
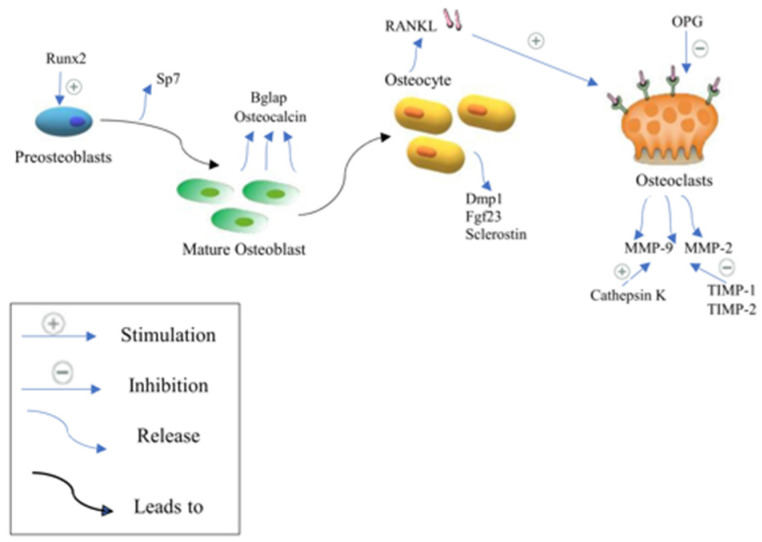
RANK/RANKL/OPG signaling pathway.

**Figure 3 biomedicines-12-01635-f003:**
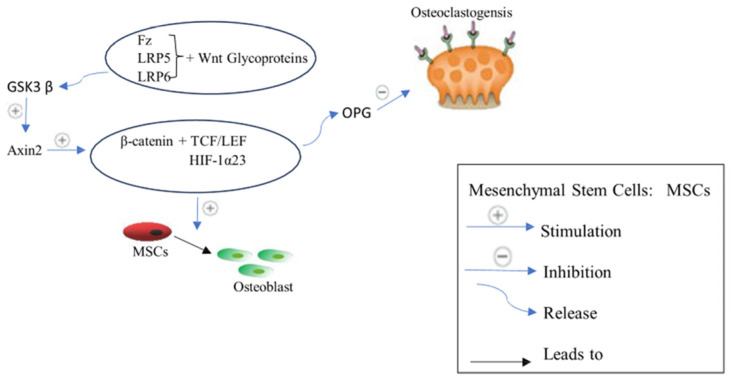
Wnt/β-catenin signaling pathway.

**Figure 4 biomedicines-12-01635-f004:**
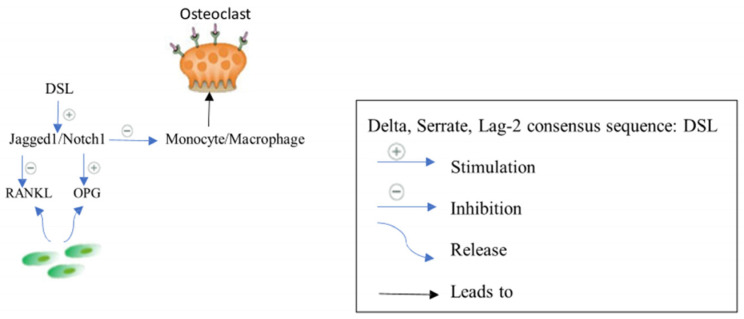
Jagged1/Notch1 signaling pathway.

**Figure 5 biomedicines-12-01635-f005:**
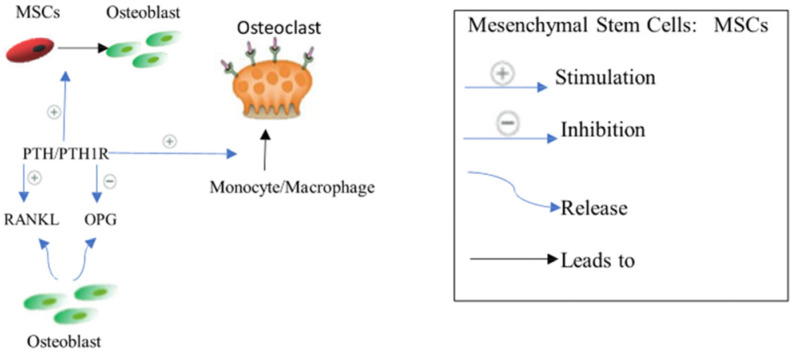
Parathyroid hormone PTH signaling pathway.

**Figure 6 biomedicines-12-01635-f006:**
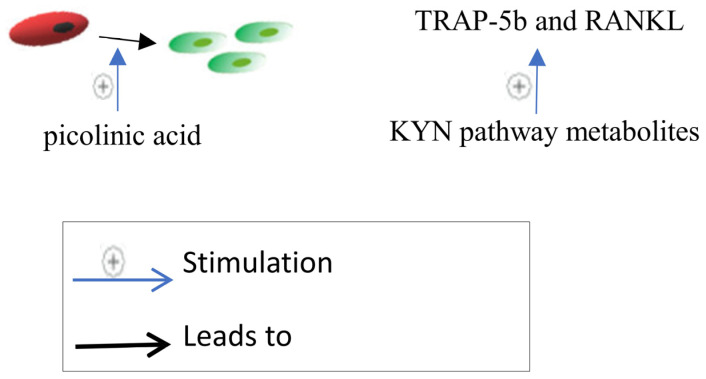
Kynurenine (KYN) pathway.

**Figure 7 biomedicines-12-01635-f007:**
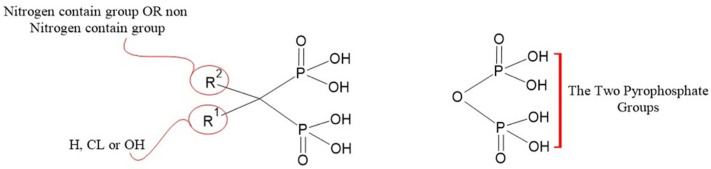
Bisphosphonates’ chemical structure.

**Figure 8 biomedicines-12-01635-f008:**
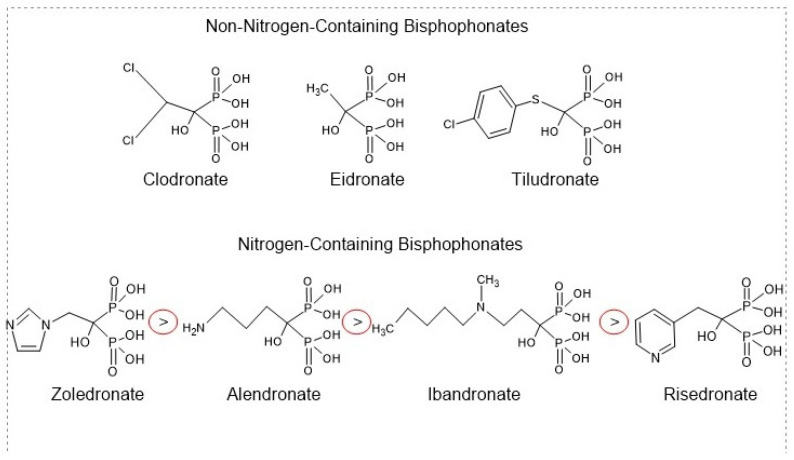
Bisphosphonates: nitrogen-containing and non-nitrogen-containing agents.

**Figure 9 biomedicines-12-01635-f009:**
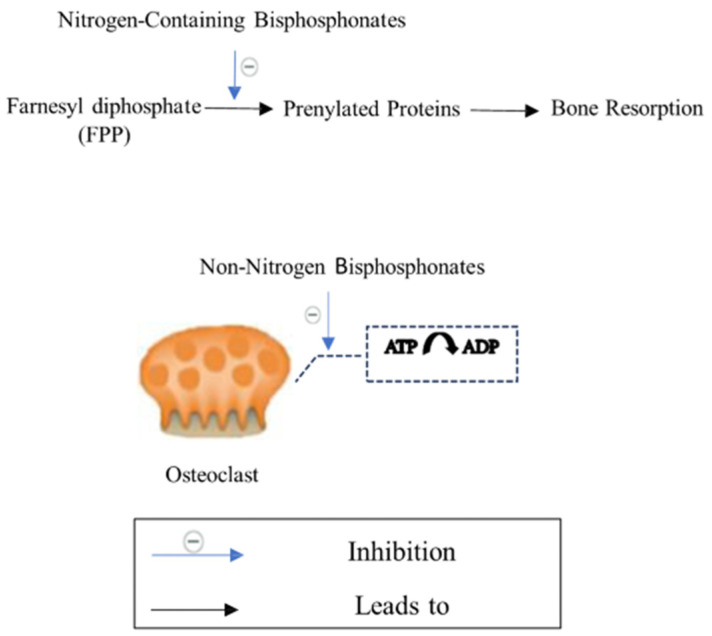
Mechanism of action of nitrogen-containing bisphosphonates.

**Figure 10 biomedicines-12-01635-f010:**
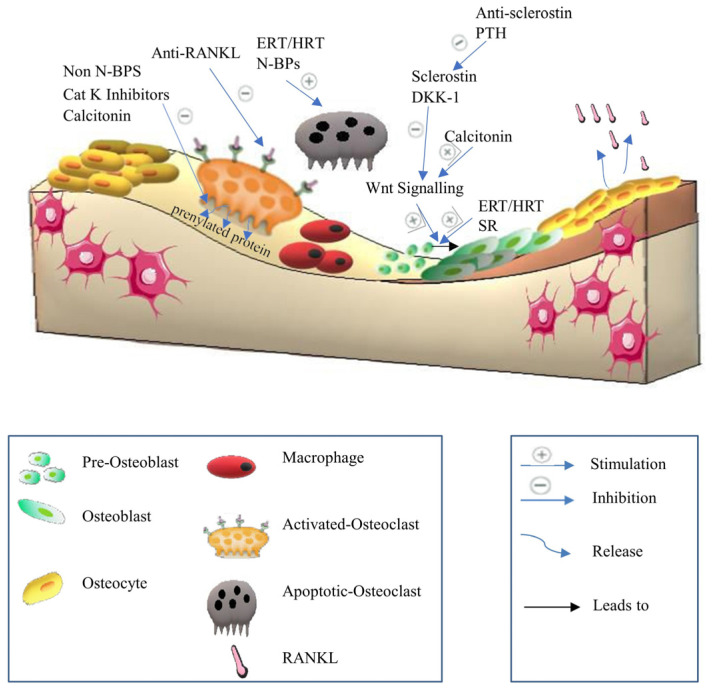
Illustrates the integration between crucial pathways and pharmacological therapies of osteoporosis and their mechanism of action. PTH: parathyroid hormone, SR: strontium ranelate, ERT/HRT: estrogen replacement therapy/hormone replacement therapy, BPs: bisphosphonates.

**Table 1 biomedicines-12-01635-t001:** The present osteoporosis medications, the mechanism of action, and side effects.

Agent	Drug Name	Drug Class	Mechanism of Action	Side Effects
Bisphosphonates	Ibandronate Alendronate Zoledronic acid Risedronate Clodronate Etidronate Tiludronate [64,65]	Anti-resorptive drug [62]	Induces osteoclast apoptosis Inhibits protons release from osteoclasts [36].	Oral administration causes dysphagia, nausea, abdominal pain, constipation or diarrhea, flatulence, acid regurgitation, esophageal ulcers, taste distortion, and gastritis. Rare side effects such as osteonecrosis of the jaw (ONJ), atypical femoral fractures (AFFs), influenza-like symptoms, uveitis, hypocalcemia, and episcleritis [69,70,71,72,73].
ERT/HRT	Estrogen/progestin along with estrogen	Anti-resorptive drug [75]	Induces osteoclast apoptosis. Decreases apoptosis in osteoblasts and osteocytes. Inhibits RANKL and boosts the synthesis of OPG. Stimulates Wnt/β-catenin signaling [37,75,76,77,78].	ERT increases the chance of cerebrovascular accidents, endometrial cancer, and venous thromboembolic disorders. HRT increases the risk of cerebrovascular accidents, cardiovascular diseases, breast cancer, and venous thromboembolic disorders [72].
SERMs	Raloxifene Bazedoxifene [12]	Anti-resorptive drug [38]	Induces osteoclast apoptosis. Decreases apoptosis in osteoblasts and osteocytes. Inhibits RANKL and boosts the synthesis of OPG. Stimulates Wnt/β-catenin signaling [38].	SERMs raise the risk of thromboembolic disorder, muscular spasms, and cerebrovascular accident [88].
Anti-RANKL antibody	Denosumab [96]	Anti-resorptive drug [95]	Blocks RANKL pathway [96].	Rebound enhancement in osteoclastogenesis, atypical femoral fractures (AFFs), osteonecrosis of the jaw (ONJ), musculoskeletal discomfort, hypocalcemia, gastrointestinal problems, and impairs the immune system in long-term use [38,98].
Cathepsin K inhibitors	Odanacatib Balicatib [107,110]	Anti-resorptive drug [106]	Enlarges osteoclasts and reduces collagen degradation via cathepsin K inhibition [106,109,110].	AFF, pycnodysostosis, and a higher risk of cerebrovascular accident [38].
Strontium ranelate	Ranelic acid Distrontium salt [134]	Anabolic and anti-resorptive drug [134]	Stimulates the ERK signaling improves osteoblast proliferation, and AKT signaling reduces osteoblast apoptosis. Stimulates IGF-1 and IGF-2, causing stimulation of osteoblast proliferation and differentiation. Activates PLC and NF-κB, causing osteoclast apoptosis. Activates OPG. Increasing osteoclast apoptosis and osteoblast proliferation and differentiation directly [128,129,130,131,132,133].	Venous thromboembolism, cardiovascular disorders, symptoms of the nervous system, and myocardial infarction, including allergic reactions like systemic symptoms syndrome and drug rash with eosinophilia [136].
Vitamin K2	Vitamin K2	Nutrition intake Anabolic effect [40]	Aids γ-carboxylation of osteocalcin, which is released via osteoblasts to form bone matrix [40].	Safe [158].
Calcitonin	Miacalcin [18]	Anti-resorptive drug [90]	Increases calcium uptake in the bone. Stimulates Wnt10b in osteoclasts, leading to bone formation induction. Limits osteoclast motility and capacity to resorb bone via transcription regulation. Inhibits the maturation of osteoclast precursors, leading to inhibit bone loss [18,89,90].	Hypocalcemia, loss of appetite, abdominal pain, nausea, diarrhea, and increasing incidence of prostate cancer [38,93].
Anti-sclerostin antibody	Romosozumab [125]	Anabolic drug [112]	Activates the canonical Wnt signaling pathway, increases bone formation, and reduces bone resorption. Suppresses the BMP7 signaling [112,122].	Myocardial infarction, cerebrovascular accidents, cardiovascular events, and malignant tumors [122,124,126,127].
PTH	Teriparatide Abaloparatide [116]	Anabolic drug [47]	Inhibits sclerostin and stimulates Wnt signaling pathway. Promotes osteoblast proliferation and differentiation [18,112,113,114,115].	Osteosarcoma risk, cephalgia, dizziness, limb cramps, nausea, and hypercalcemia [119,120].
Calcium	Calcium	Nutrition intake Anti-resorptive drug [149]	Lower PTH secretion [149].	Gastrointestinal disorders and hypercalcemia [149].
Vitamin D	Vitamin D	Nutrition intake Anti-resorptive drug [154]	Promotes the intestinal intake of calcium [154].	Hypercalcemia, gastrointestinal adverse effects, renal calculus, and myocardial infarction, which exceed the limited advantages of the therapy [156].
TNF-α inhibitors	Infliximab [138]	Anti-resorptive drug [137]	Leads to serum level changes for bone resorption and formation markers, but further studies are needed [138,139,140].	Congestive heart failure and aplastic anemia, low tissue penetrating capacity, and oral availability [146,147,148].
IL-6 inhibitors	Tocilizumab [141]	Anti-resorptive drug [141]	Stimulates PINP, which is a bone formation marker, and inhibits CTX-1 and ICTP, which are bone resorption markers [141,142,143].	Congestive heart failure and aplastic anemia, low tissue penetrating capacity, and oral availability [146,147,148].
Stem cells	Stem cells	Source of osteoblasts [165]	MSCs directly cover the affected area in bones and differentiate into osteoblasts. MSCs secrete a variety of growth factors that inhibit osteoclastic differentiation and promote angiogenesis indirectly [165,166].	Not well documented [167].

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
