# Peer review of "Mechanistic Insights and Therapeutic Strategies in Osteoporosis: A Comprehensive Review"

_biomedicines, 2024, doi:10.3390/biomedicines12081635_

Round 1

Reviewer 1 Report

Comments and Suggestions for Authors

In this interesting review by N. Muhammad and co-workers, the authors highlight the mechanisms involved and therapeutic strategies used to treat Osteoporosis. The review is quite well written, and the work significant to the field, however I recommend revisions prior publication in biomedicines.

The first sections (i.e., 2.1 to 3.1.2) are very difficult to understand, and extrapolate. Schemes should be added to help the reader or at least Fig. 2 should be improved.

L2: Title (??) Mechanistic…..

L39: minus sign must be inserted via the insert symbol command.

L49‒L54: the same paragraph has been written twice.

Figure 1 and 2 are of low quality.

L216: proteins that have been found?.

L265: space after PTH and before is.

L296: hyphen after by and before product.

L356: sentence to reconsider. Patients must fast?

L531‒543: same paragraph written twice.

L717: remove a dot after patients.

Reference section: please use the good template for references.

Comments on the Quality of English Language

A Minor editing of the English language is required.

Author Response

Thank you very much for reviewing our manuscript. Below are the responses (R) to your comments:

  1. The first sections (i.e., 2.1 to 3.1.2) are very difficult to understand and extrapolate. Schemes should be added to help the reader or at least Fig. 2 should be improved.

R: We have restructured some of the sentences and added schematic diagrams to illustrate our points.

  1. L2: Title(??) Mechanistic…..

R: We apologize for the mistake. We have removed the word “title’

  1. L39: minus sign must be inserted via the insert symbol command.

R:  We have changed it to ≤from the symbol command.

  1. L49‒L54: the same paragraph has been written twice.

R: We have removed the duplicates. Our apologies for the mistake.

  1. Figure 1 and 2 are of low quality.

R: We have amended the figures to a better quality.

  1. L216: proteins that have been found?.

R: We edited the sentence to ‘Various human Wnt proteins that have been identified’.

  1. L265: space after PTH and before is.

R: We have amended this (L275)

  1. L296: hyphen after by and before product.

R: We have amended this (L307)

  1. L356: sentence to reconsider. Patients must fast?

R: We have revised the sentence to avoid confusion (L376-378)

  1. L531‒543: same paragraph written twice.

R: We have amended this

  1. L717: remove a dot after patients.

R: We have amended this

  1. Reference section: please use the good template for references

R: We have corrected this.

A Minor editing of the English language is required.

R: We have made the corrections throughout the manuscript.

Reviewer 2 Report

Comments and Suggestions for Authors

Dear Editor and Authors,

I would like to express my gratitude for the opportunity to review the manuscript titled "Mechanistic Insights and Therapeutic Strategies in Osteoporosis: A Comprehensive Review." It is a well-written and interesting paper that provides a thorough examination of osteoporosis, its pathophysiological mechanisms, and various therapeutic interventions. The manuscript includes all relevant information about osteoporosis and types of interventions, and is well-structured. The figures provided are correct and greatly aid in understanding the mechanisms discussed.

However, I have only a few suggestions to improve the manuscript. 

I would like to ask the authors to consider adding references in Table 1. Providing references for the information presented in the table would enhance the credibility and thoroughness of the review.

I suggest changing those non-MeshTerms keywords to MeshTerms

Comments on the Quality of English Language

None, seems well written.

Author Response

However, I have only a few suggestions to improve the manuscript. 

I would like to ask the authors to consider adding references in Table 1. Providing references for the information presented in the table would enhance the credibility and thoroughness of the review.

R: We have added the references as suggested

I suggest changing those non-MeshTerms keywords to MeshTerms

 R: We have changed the keywords to Mesh Terms.

Round 2

Reviewer 1 Report

Comments and Suggestions for Authors

In this very interesting work by N. Muhammad and co-workers, the authors highlight the mechanisms involved and the therapeutic strategies used to treat Osteoporosis. The review is very well written, and the work significant to the field. The authors have performed all the recommended modifications going even further, so I recommend publication in biomedicines after the following minor revisions have been carried out.

L12: patients’s quality of life

Figure 1: The first scheme might be removed to prevent redundancy with the second.

Reference section: please use the good template for references.

Fig. 2,7,8,10 are of low quality.

L216: Missing a word before also play.

L303: F of figure 2 = capital letter

L345: bonds of P-C-P must be inserted via the insert symbol command (not hyphen).

L347: osteoclasts’s

L732: …in a several clinical studies. text to be restructured.

Fig. 10. The two schemes may be redundant.

Table 1: Side effects of stem cells = safe. Safe should be taken with a grain of salt

References section: use the good template. See MDPI Reference List and Citations Style Guide at https://www.mdpi.com/authors/references

Journals name must be abbreviated.

Comments on the Quality of English Language

The English language is fine. No issues detected.

Author Response

Thank you very much for taking the time to review our manuscript. Below are the responses (R) to your comment:

L12: patients’s quality of life.

R: We have corrected this (L17)

Figure 1: The first scheme might be removed to prevent redundancy with the second.

R: We have removed Fig 1 as suggested.

Reference section: please use the good template for references.

R: We have updated the reference using the template in the Endnote.

Fig. 2,7,8,10 are of low quality.

R: We have converted all the figures to 600dpi and pasted them into the manuscript. However, we were not able to upload the figures in JPEG/TIFF format as the journal system only allows 1 file to be uploaded.

L216: Missing a word before also play.

R: We have corrected this (L207)

L303: F of figure 2 = capital letter

R: We have corrected this (L291)

L345: bonds of P-C-P must be inserted via the insert symbol command (not hyphen).

R: We have corrected this (L332)

L347: osteoclasts’s

R: We have corrected this (L334)

L732: …in a several clinical studies. text to be restructured.

R: We have revised the sentence (L694)

Fig. 10. The two schemes may be redundant.

R: We have removed the first scheme.

Table 1: Side effects of stem cells = safe. Safe should be taken with a grain of salt

R: Yes, we agree. We changed this to 'Not-well documented'

References section: use the good template. See MDPI Reference List and Citations Style Guide at https://www.mdpi.com/authors/references

Journals name must be abbreviated.

R: We have used the template and downloaded it in the EndNote. However, we were not able to set the journal names in the abbreviated version. We understand that the Editorial Office will do the abbreviation for us (as stated in the Instructions to Authors section) 
